
# Fluctuations of work in realistic equilibrium states of quantum systems with conserved quantities

**Jordi Mur-Petit[1*], Armando Relaño[2], Rafael A. Molina[3] and Dieter Jaksch[1,4]**

**1** Clarendon Laboratory, University of Oxford, Parks Road, Oxford OX1 3PU, United Kingdom
**2** Departamento de Estructura de la Materia, Física Térmica y Electrónica, and GISC, Universidad Complutense de Madrid, Av. Complutense s/n, 28040 Madrid, Spain
**3** Instituto de Estructura de la Materia, IEM-CSIC, Serrano 123, 28006 Madrid, Spain
**4** Centre for Quantum Technologies, National University of Singapore, 3 Science Drive 2, 117543 Singapore

⋆ jordi.murpetit@physics.ox.ac.uk

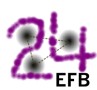
## Abstract

The out-of-equilibrium dynamics of quantum systems is one of the most fascinating problems in physics, with outstanding open questions on issues such as relaxation to equilibrium. An area of particular interest concerns few-body systems, where quantum and thermal fluctuations are expected to be especially relevant. In this contribution, we present numerical results demonstrating the impact of conserved quantities (or 'charges') in the outcomes of out-of-equilibrium measurements starting from realistic equilibrium states on a few-body system implementing the Dicke model.

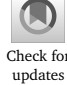
## 1 Introduction

Understanding how a generic (many-body) physical system evolves in time from an arbitrary initial state and relaxes (or not) to an equilibrium state is a fundamental problem underlying questions from the cooling of neutron stars [1,2] to the design of materials that quickly remove excess heat from computing chips in cell phones [3,4].

In classical physics, conservation laws (e.g., on energy, momentum, angular momentum) can severely constrain the phase space available to the system, thus enabling to make precise predictions on some of these questions. In quantum physics, conservation laws play a similarly strong role. This was strikingly demonstrated in the quantum Newton's cradle experiment [5]. In this experiment, a one-dimensional (1D) gas of strongly-interacting bosons in a harmonic trap was initialized in a highly-non-equilibrium state, and observed not to relax even after a long time evolution (hundreds of trap periods, which sets the natural timescale of the problem

and imply thousands of atomic collisions). This behaviour is understood by noting that the systems, in the limit of infinitely-strong interactions, is best described as a Tonks-Girardeau gas [6,7], which is an an integrable model, i.e., it features an extensive number of *conserved charges*. These are operators, $\hat{M}_k$, that commute with the system's Hamiltonian, $[\hat{H},\hat{M}_k]=0$ ($k=1,\ldots,N_{\mathrm{cons}}$). In this limit of strong interactions, one can calcualte the expectation values of few-body observables after relaxation by describing the relaxed state of the system by a generalization of the Gibbs ensemble (GGE) [8], see Eq. (1). In the conditions of isolation in which the experiment occurs, the system is unable to change the value of these charges, which effectively precludes relaxation to a Gibbs equilibrium state [5,8].

More recently, Schmiedmayer *et al.* have presented a series of experiments on similar 1D Bose gases [9–12] (see also [13,14]). By subjecting the system to quenches, they explored the emergence, at long but intermediate timescales, of a (pre-)thermalised state, which is determined by the values of the conserved charges at the start of the evolution. These experiments brought to light the need to include information on the charges in the description of the *equilibrium* state of a quantum many-body system, when the system's Hamiltonian supports them. These findings are in agreement with very general theoretical principles from quantum thermodynamics [8,15,16], that demand that the equilibrium state of such a system be described by a density matrix of the form of the generalised Gibbs ensemble (GGE), namely

$$\rho_{\mathrm{GGE}} = \exp\left(-\beta\hat{H} - \sum_k \beta_k \hat{M}_k\right)/Z_{\mathrm{GGE}}\,, \tag{1}$$

$$Z_{\mathrm{GGE}} \equiv Z_{\mathrm{GGE}}(\hat{H},\beta,\{\hat{M}_k,\beta_k\}) = \mathrm{tr}\left[\exp\left(-\beta\hat{H} - \sum_k \beta_k \hat{M}_k\right)\right]. \tag{2}$$

Here, $\beta$ is the usual inverse temperature, while $\{\beta_k\}(k=1,\ldots,N_{\mathrm{cons}})$ are called generalised inverse temperatures.

The fact that the equilibrium state is of the GGE form has implications for the expectation values of measurements done on the system in equilibrium, as has been extensively analysed with numerical simulations on a range of models [8,17–21]. It is more difficult to make generic statements on the implications of the charges on *non-equilibrium* measurements of a quantum many-body system. A milestone result in classical non-equilibrium thermodynamics is the discovery of exact relations between equilibrium and non-equilibrium measurements, starting with the theorems on the large fluctuations of entropy production in fluids under shear stress [22–24], and including the Jarzynski equation between work and free energy [25].

Several authors have derived analogous relations, dubbed *quantum fluctuations relations* (QFRs), for closed quantum systems, assuming their state at the start of the process is of the standard Gibbs form:

$$\rho_{\mathrm{Gibbs}} = \exp\left(-\beta\hat{H}\right)/Z, \quad Z \equiv Z(\hat{H},\beta) = \mathrm{tr}\left[\exp\left(-\beta\hat{H}\right)\right]. \tag{3}$$

More recently, the present authors have generalised these QFRs to the case that the equilibrium state is of the GGE form and for an arbitrary number of charges for the initial and final states, thus notably expanding the rage of non-equilibrium problems that can be tackled [26]. In particular, our formalism is explicitly able to deal with processes where the number of charges of the initial and final Hamiltonians differ (cf. [27]), and thus enables one to address fundamental open questions on the thermalization of integrable systems when perturbed away from integrability [5,9,11,28,29].

An important question that remained unanswered in [26] was: how sensitive are the generalised QFRs to the initial state not being a perfect GGE? In other words: if we have a system with charges, and can only generate an imperfect equilibrium state that is only approximately given by Eq. (1), will non-equilibrium measurements be able to distinguish this from a 'simple'

Gibbs state (3)? In this contribution, we provide numerical evidence supporting an affirmative answer to this question.

## 2 Review of generalized quantum fluctuation relations

We start by briefly reviewing the main results in Ref. [26], in particular the generalised versions of the quantum Jarzynski [30–32] and Tasaki-Crooks [33] relations. In analogy to the derivations of the standard QFRs [30–33], we consider an initial equilibrium state. In agreement with Jaynes' information-theory formulation of statistical mechanics, if the Hamiltonian features some charges $\hat{M}_k$, this initial equilibrium state will be of the GGE form (1), with the set of generalised inverse temperatures $\vec{\beta} = \{\beta, \{\beta_k\}\}$ determined by requiring that the following equalities on expectation values are satisfied:

$$\mathrm{tr}\big[\rho_{\mathrm{GGE}}(\vec{\beta})\hat{H}\big] = \overline{E} \tag{4}$$

$$\mathrm{tr}\big[\rho_{\mathrm{GGE}}(\vec{\beta})\hat{M}_k\big] = \overline{M_k}, \quad k = 1, \ldots, N_{\mathrm{cons}}. \tag{5}$$

Here, $\overline{E}$ is the energy of the initial state, and $\overline{M_k}$ the expectation value of operator $\hat{M}_k$ in the initial state.

We then submit the system to an out-of-equilibrium process by changing its Hamiltonian from the initial value $\hat{H}$ to some new final Hamiltonian $\hat{H}'$. In general, we expect the set of operators that commute with $\hat{H}'$ to be different from that of charges of $\hat{H}$, and we label the latter $\hat{M}'_k$, $[\hat{H}', \hat{M}'_k] = 0$ ($k = 1, \ldots, N'_{\mathrm{cons}}$).

To quantify the amount of energy, and the energy fluctuations, imparted on the system by this process, we consider a generalised version of the two-energy-projection measurement (TPM) protocol [34], as introduced in [26]:

1. At time $t = 0$, we project the initial state onto the basis of eigenstates of the initial Hamiltonian, $\big|n, i_1, \ldots, i_{N_{\mathrm{cons}}}\big\rangle$, with the spectral decomposition of the Hamiltonian $\hat{H}\big|n, i_1, \ldots, i_{N_{\mathrm{cons}}}\big\rangle = E_n\big|n, i_1, \ldots, i_{N_{\mathrm{cons}}}\big\rangle$, and that for the charges, $\hat{M}_k\big|n, i_1, \ldots, i_{N_{\mathrm{cons}}}\big\rangle = M_{k,i_k}\big|n, i_1, \ldots, i_{N_{\mathrm{cons}}}\big\rangle$. In other words, $n$ stands for the quantum number that identifies the energy eigenvalue, $E_n$, while $i_k$ is the quantum number labelling the eigenvalues, $M_{k,i_k}$, of the charge operator $\hat{M}_k$. We obtain a definite value for the energy, $\mathcal{E}_{\mathrm{ini}} \in \{E_n\}$, and the other charges, $\mu_{k,\mathrm{ini}} \in \{M_{k,i_k}\}$ ($k = 1, \ldots, N_{\mathrm{cons}}$).

2. Next, we drive the system out of equilibrium by steering its Hamiltonian, $\hat{H} \mapsto \hat{H}(t)$, for times $0 < t < t_{\mathrm{fin}}$. We impose no limitation in the form of the time dependence. This driving defines a unitary time-evolution operator $U(t)$ that is the solution of $i\hbar\partial_t U(t) = \hat{H}(t)U(t)$, with $U(0) = \mathbb{I}$, the identity operator in the system's Hilbert space.

3. Finally, at time $t = t_{\mathrm{fin}}$, we project the system on the eigenbasis of the final Hamiltonian, $\hat{H}' = \hat{H}(t_{\mathrm{fin}})$, $\big|m', i'_1, \ldots, i'_{N'_{\mathrm{cons}}}\big\rangle$, with $\hat{H}'\big|m', i'_1, \ldots, i'_{N'_{\mathrm{cons}}}\big\rangle = E'_m\big|m', i'_1, \ldots, i'_{N'_{\mathrm{cons}}}\big\rangle$, and the corresponding charges, $\hat{M}'_k\big|m', i'_1, \ldots, i'_{N'_{\mathrm{cons}}}\big\rangle = M'_{k,i_k}\big|m', i'_1, \ldots, i'_{N'_{\mathrm{cons}}}\big\rangle$. This gives definite values for the final energy, $\mathcal{E}_{\mathrm{fin}} \in \{E'_m\}$, and the other charges, $\mu_{k,\mathrm{fin}} \in \{M'_{k,i_k}\}$ ($k = 1, \ldots, N'_{\mathrm{cons}}$).

Together with this 'forward' (FW) protocol, we consider a twin protocol, that starts at time $t = 0$ with the system in the GGE equilibrium state of the Hamiltonian $\hat{H}'$ and changes it into $\hat{H}$ following the time-reversed evolution, i.e., with the unitary $U^{-1}(t)$. Note that the initial state of this 'backward' (BW) protocol will have associated in general a different set of

generalised inverse temperatures, $\vec{\beta}' = \{\beta', \{\beta_k'\}\}$. We define the work, $w$, and generalised work, $\mathcal{W}$, done on the system after a single run of these protocols as:

$$w = \mathcal{E}_{\text{fin}} - \mathcal{E}_{\text{ini}}, \tag{6}$$

$$\mathcal{W} = \left( \beta' \mathcal{E}_{\text{fin}} + \sum_k \beta_k' \mu_{k,\text{fin}} \right) - \left( \beta \mathcal{E}_{\text{ini}} + \sum_k \beta_k \mu_{k,\text{ini}} \right). \tag{7}$$

These are stochastic quantities, as they depend on the result of projective measurements at the start and end of the process. The Tasaki-Crooks relation [33] is the following relationship between the probability distribution functions (PDFs) of the variable $w$ in the FW and BW processes:

$$\frac{P_{\text{FW}}(w)}{P_{\text{BW}}(-w)} e^{-\beta w} = \frac{Z(\hat{H}', \beta')}{Z(\hat{H}', \beta)} \equiv \exp(-\beta \Delta F), \tag{8}$$

where $\Delta F = Z(\hat{H}', \beta)/Z(\hat{H}, \beta)$ is the difference in free energies between the two *equilibrium* states, with the partition functions defined as in Eq. (3). By multiplying both sides of (8) by $P_{\text{BW}}(-w)$ and integrating over $w$ one retrieves the quantum Jarzynski equality [30–32]:

$$\langle \exp(-\beta w) \rangle = \exp(-\beta \Delta F), \tag{9}$$

where $\langle \cdot \rangle$ stands for an average over many runs of the protocol. Eqs. (8) and (9) hold when the initial state is of the form of a Gibbs state, Eq. (3).

In Ref. [26] we have shown that when the initial state is of the form of the GGE form, Eq. (3), the PDF of of generalised work, $\mathcal{W}$, satisfies instead a generalised Tasaki-Crooks relation that reads:

$$\frac{P_{\text{FW}}(\mathcal{W})}{P_{\text{BW}}(-\mathcal{W})} e^{-\mathcal{W}} = \frac{Z_{\text{GGE}}(\hat{H}', \beta', \hat{M}_k', \beta_k')}{Z_{\text{GGE}}(\hat{H}, \beta, \hat{M}_k, \beta_k)} \equiv \exp(-\Delta \mathcal{F}), \tag{10}$$

with the partition functions in the GGE, $Z_{\text{GGE}}$, defined in (1), and $\Delta \mathcal{F} = \mathcal{F}' - \mathcal{F}$ the difference in generalised (dimensionless) free energy functions, $\mathcal{F} = -\ln Z_{\text{GGE}}$ and $\mathcal{F}' = -\ln Z_{\text{GGE}}'$. Analogously to above, if we multiply both sides of Eq. (10) by $\mathcal{P}_{\text{BW}}(-\mathcal{W})$ and integrate over $\mathcal{W}$, we obtain the following equality:

$$\langle \exp(-\mathcal{W}) \rangle = \exp(-\Delta \mathcal{F}). \tag{11}$$

This is the generalised quantum Jarzynski equality [26].

# 3 Testing the generalized QFRs with an imperfect GGE

## 3.1 Dicke model

In Ref. [26] we presented extensive numerical results testing both the standard, Eqs. (8) and (9), and generalised QFRS, Eqs. (10) and (11). We found that when the initial state of either one or both initial equilibrium states in the FW and BW processes is not of the Gibbs form but a GGE, the standard relations fail, while the generalised ones are satisfied perfectly.

Here, we consider a more general question, which is to what extent it is necessary for the system to be in a perfect GGE equilibrium state for the generalised QFRs to provide a good prediction for the statistics of generalised work in out-of-equilibrium processes.

To this end, following Ref. [26], we consider a system composed of $N$ two-level systems, with energy splitting $\omega_{\text{at}}$, coupled with equal strength $g$ to a bosonic field of frequency $\omega_{\text{b}}$,

i.e., the $N$-particle Dicke model [35–37]. We write the Hamiltonian describing this system in the form [38–41]:

$$H = \hbar\omega_{\text{b}}\hat{b}^{\dagger}\hat{b} + \hbar\omega_{\text{at}}\hat{J}_z + \frac{2g}{\sqrt{N}}\Big[(1-\alpha)\big(\hat{J}_+\hat{b} + \hat{J}_-\hat{b}^{\dagger}\big) + \alpha\big(\hat{J}_+\hat{b}^{\dagger} + \hat{J}_-\hat{b}\big)\Big], \qquad (12)$$

where $\hat{b}^{\dagger}$ and $\hat{b}$ are the operators creating and annihilating excitations in the bosonic field, and $\hat{J}_s$ ($s = z, +, -$) are Schwinger spin operators describing the collective internal state of the two-level systems, with $J = N/2$. This model was introduced to describe the coupling of atoms to light fields [35]. More recently, it has been implemented in systems of trapped ions [41].

In Eq. (12) we have introduced $g$, the coupling strength between two-level systems and the boson field, and the parameter $0 \le \alpha \le 1$. When $\alpha = 0$ or $\alpha = 1$, the Dicke Hamiltonian reduces to the Tavis-Cummings model, which is integrable and has an additional conserved quantity, the total number of excitations in the system, $\hat{M} = \hat{J} + \hat{J}_z + \hat{b}^{\dagger}\hat{b}$; otherwise, for $0 < \alpha < 1$, the model is in the chaotic regime [26, 38–40, 42]. Thus, we can analyse the behaviour of this system in the integrable and non-integrable limits simply by considering cases with $\alpha \in \{0, 1\}$ and $\alpha \notin \{0, 1\}$, respectively. In Ref. [26] we have discussed how this tuning can be accomplished in trapped-ion setups by controlling the intensity of the light fields implementing the red- and blue-sideband transitions with respect to the centre-of-mass mode, that plays the role of the bosonic field, $\hat{b}$.

## 3.2  Numerical results

Our numerical studies testing the standard and generalised QFRS in Ref. [26] were obtained assuming that the system is initially equilibrated, and hence perfectly described by either a Gibbs, with inverse temperature $\beta$, or a GGE density matrix, with two generalised temperatures, $\beta$ and $\beta_M$. A recent work by one of us [42] shows that the usual concept of thermalisation —the equivalence between microcanonical ensemble and long-time averages of physical observables— is not always enough to guarantee the applicability of standard quantum fluctuation relations. Here, we show that our generalised QFRs are robust and provide a good description of non-equilibrium processes starting from real equilibrium states in integrable systems.

To tackle this question we design the following protocol:

1. We start from a thermal Gibbs state, with $\beta = 0.02$, in a chaotic configuration of the Dicke model, with $\alpha = 1/2$ and $g = \epsilon_0$, being $\epsilon_0$ the energy scale of the problem.[1]

2. We perform the forward protocol directly quenching the system onto an integrable configuration, with $\alpha = 0$ and $g = 6\epsilon_0$, i.e., the time-dependence of the Hamiltonian parameters reads

$$\alpha(t) = \begin{cases} 1/2 & t < 0 \\ 0 & t \ge 0 \end{cases}, \qquad \text{and} \qquad g(t) = \begin{cases} \epsilon_0 & t < 0 \\ 6\epsilon_0 & t \ge 0 \end{cases}.$$

We emphasize that our generalised QFRs do not depend on this specific choice of time dependence, and we have chosen it for computational convenience. Other variations of $(\alpha, g)$ with the same initial and final values would render the same results on the left- and right-hand sides of Eqs. (10) and (11), see [26].

---

[1]In our numerical calculations, we set $N = 7$, $\hbar\omega_{\text{b}} = 3\epsilon_0$, $\hbar\omega_{\text{at}} = 10\epsilon_0$, and include up to $n = 800$ in the bosonic field. As the dimension of the bosonic Hilbert space is actually infinite, this high number has been chosen to guarantee that all the Fock states with non-zero occupation probability are included in our simulations. In an experimental implementation of the Dicke model with trapped ions [26, 41, 43], the energy scale can be fixed to be of the order of the trapping frequency, $\epsilon_0 = h \times 1$ MHz (with $h$ Planck's constant) [41, 43–45].

3. We perform the backward protocol from the resulting state[2].

We calculate statistics of work for the forward process —i.e., the PDFs $P_{\mathrm{FW}}(w)$ and $P_{\mathrm{FW}}(\mathcal{W})$— from steps 1-2, and for the backward process from steps 2-3. We compare these statistics of work with two reference distributions: a GGE with the values $\beta$ and $\beta_M$ obtained from least-square fits *of the actual time-evolved state after step 2* to the expected values of $\langle \hat{H} \rangle$ and $\langle \hat{M} \rangle$; and a standard Gibbs ensemble, with $\beta$ obtained from a least-square fit to the expected value of $\langle \hat{H} \rangle$.

It is worth noting that this protocol challenges our QFRs in the most demanding scenario. When describing the initial equilibrium state by means of a GGE, both the number of conserved charges and the values of the generalised temperatures are different from the ones in the state from which the forward protocol starts. In the other case, when a standard Gibbs ensemble is taken as a reference, the number of charges is the same —just the Hamiltonian itself—, but the values of the temperatures are different.

Results are summarized in Fig. 1. Panels (a) and (c) show that the equilibrium state after the forward protocol is pretty well described by means of a GGE with $\beta = 2.76 \cdot 10^{-3}$ and $\beta_M = 1.41 \cdot 10^{-1}$ (see the caption for more details), and poorly described by means of a standard Gibbs ensemble with $\beta = 6.02 \cdot 10^{-3}$. As the quench ends in an integrable configuration, the role of the conserved charge $\hat{M}$ is essential to properly describe the equilibrium state.

Fig. 1(b) and (d) summarize the results testing the Tasaki-Crooks relation and its generalised version. Fig. 1(b) shows that the generalised version, Eq. (10), accounts for the statistics of the generalised work, $\mathcal{W}$, with high precision. Only two points around $\mathcal{W} \approx 1.2$ are overestimated by the formula. This reinforces the former conclusion stating that the GGE provides a very accurate picture of the state after the forward part of the protocol. Our results point out that this is true, not only for expectation values of physical observables in equilibrium, but also for the statistics of work and other conserved charges in non-equilibrium processes.

In contrast to this, Fig. 1(d) clearly shows that the standard version of the Tasaki-Crooks relation, Eq. (8), fails to account for the statistics of work. This fact is directly linked to the results shown in Fig. 1(c): As the occupation probabilities after the forward part of the protocol are not well described by a standard Gibbs ensemble, the statistics of work resulting from such a state does not follow the standard Tasaki-Crooks relation.

## 4   Conclusion

In summary, we have presented generalized versions of the Tasaki-Crooks and Jarzynski quantum fluctuation relations, that are suitable to study the out-of-equilibrium dynamics of systems with an arbitrary, possibly time-dependent, number of charges [26]. These exact relations assume that the state of the quantum system at the start of the out-of-equilibrium process is of the form of the generalized Gibbs ensemble, in accordance with very general principles of quantum statistical mechanics.

In this contribution, we have tested the validity of our generalised QFRs [26] to a more stringent test by considering a more realistic situation, in which the system is not allowed an infinite time to relax to its equilibrium state in contact to baths. Our robust numerical calculations support that, when the Hamiltonian describing the system has conserved charges,

---

[2]To be sure that we start from an equilibrium state, we must let the system relax in the final Hamiltonian, $\alpha = 0$ and $g = 6\epsilon_0$, before starting the backward part of the protocol. However, this relaxation time is irrelevant for our numerical simulation. All our results are based on the two-projective measurement scheme. Hence, if the actual state of the system at a certain time $t$ is $|\Psi(t)\rangle = \sum_n C_n(t)|\Phi_n\rangle$, where $|\Phi_n\rangle$ are the eigenfunction of the Hamiltonian, only the square moduli of the coefficents, $|C_n|^2$, are relevant. Therefore, the dephasing introduced by the relaxation procedure does not play any role in the results.

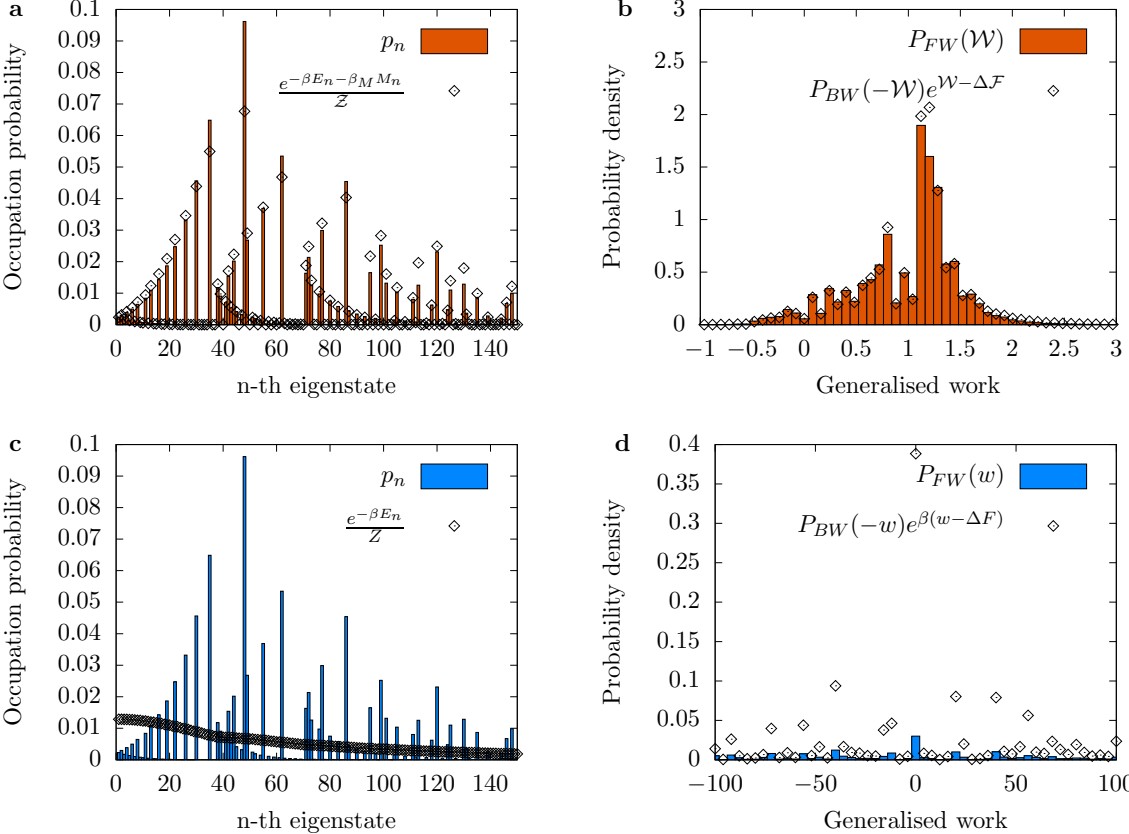

Figure 1: Panels (a) and (c) compare the numerical results for the occupation numbers in the state after the forward quench (solid histograms) with the reference distributions (diamonds). In panel (a), the reference distribution is a GGE with $\beta = 2.76 \cdot 10^{-3}$ and $\beta_M = 1.41 \cdot 10^{-1}$ (values obtained from a least-square fit to the values of $\langle \hat{H} \rangle$ and $\langle \hat{M} \rangle$). In panel (c), the reference distribution is a standard Gibbs with $\beta = 6.02 \cdot 10^{-3}$ (value obtained from a least-square fit to the value of $\langle \hat{H} \rangle$). Panels (b) and (d) show the results for the (generalised version of) the Tasaki-Crooks theorem. Results for the forward distributions are displayed with solid histograms, and results for the backwards, together with the factors $e^{\mathcal{W} - \Delta \mathcal{F}}$ or $e^{\beta(w - \Delta F)}$, with diamonds. Panel (b) refers to the GGE case, and panel (d) to the standard Gibbs ensemble.

the statistics of work produced by a non-equilibrium process that starts from such a realistic equilibrium state cannot be described by using the standard QFRs (which disregard the effect of charges). On the contrary, work statistics is accurately described by our generalised QFRs, Eqs. (10)-(11). This points to the importance of the role of charges in realistic non-equilibrium processes, such as equilibration in quasi-integrable systems [28], and dissipation and relaxation in driven systems with conservation laws [46,47]. A case of particular theoretical interest for future exploration arises when the charges supported by the Hamiltonian do not commute with each other [29,48–51]. Our results also call attention to the relevance of charges in the work statistics of realistic cyclic processes where the system is driven to an intermediate state with charges, an issue that may be exploited to design more efficient quantum engines [52–55].

## Acknowledgements

We acknowledge useful discussions with B. Buča, J. Dukelsky, J. J. García-Ripoll, D. Lucas, and K. Thirumalai.

**Funding information** This work was supported by EPSRC Grant No. EP/P01058X/1 (QSUM), EU H2020 Collaborative project QuProCS (Grant Agreement No. 641277), the European Research Council under the EU's Seventh Framework Programme (FP7/2007-2013)/ERC Grant Agreement No. 319286 (Q-MAC), Spain's Grants Nos. FIS2015-63770-P (MINECO/FEDER) and PGC2018-094180-B-I100 (MCIU/AEI/FEDER, EU), CAM/FEDER Project No. S2018/TCS-4342 (QUITEMAD-CM) and CSIC Research Platform PTI-001.

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
