# Peer review of "Fluctuations of work in realistic equilibrium states of quantum systems with conserved quantities"

_SciPost Physics Proceedings, doi:SciPost Phys. Proc. 3, 024 (2020)_

## Round 1 · Referee Report · Arnau Rios Huguet (Referee 1) · 2019-12-4

Strengths

  1. Provides new results that are insightful in the context of equilibration in quantum systems
  2. Describes well work done by authors and other in the past

Weaknesses

  1. Some very minor issues could be discussed in more detail - but length was presumably a concern in the original write-up

Report

The manuscript "Fluctuations of work in realistic equilibrium states of quantum systems with conserved quantities" by Mur-Petit, Relaño, Molina and Jaksch is an interesting contribution that discusses the interplay between conserved charges, dynamics and statistical ensembles - all relevant topics in thermalization of few-body quantum states. In addition to providing a summary of previous results, this proceedings contribution expands on previous works from the authors (Ref [22]) using new simulations that challenge some of the previous existing assumptions. On this basis alone, and the quality of the manuscript, I would normally be happy to accept this contribution to SciPost for the 24th European Few Body Conference. Having said that, I would like to ask clarification of some very minor issues before the paper is fully accepted.

Requested changes

The issues are listed below in order of appearance in the manuscript and I am sure the authors will be able to tackle them quickly in a new iteration of the manuscript:

1) Lines 23-24: "cooling of neutron stars" and "excess heat from chips": these two examples of thermalization are very specific. Can the author provide references that link these to the problem discussed here?

2) Notation in the protocol described in lines 93-104: I would have initially envisaged that the eigenstates of the system were defined not only by the quantum number associated to H, |n>, but also by the quantum numbers of the operators that commute with it - eg |n, k_1, k_2 ... k_{N'_cons} > . Are the k_i labels suppressed here for simplicity? I suppose that upon projecting on the energy, without specifying M_k, the eigenstates |n> will be a mixture of the different k_i eigenstates, which are then further distilled in each projection over M_k. Is this correct?

3) Still within the protocol, the authors do not state any prescription for the driving mechanism out of equilibrium. What is the time dependence in H(t) (or, equivalently, in U(t)), while joining the initial and final states? Are there any timescales or specific shapes that could be relevant in this switching procedure? I presume continuity in t is, for instance, a prerequisitie. One also presumably needs to be in a non-adiabatic regime to guarantee a non-equilibrated final state, in the context of the extended calculations presented here?

4) Line 148: there seems to be a missing reference between "see...; " & "otherwise"

---

## Round 3 · Author Response

We very much appreciate the Referee’s positive outlook on our paper, and have modified our manuscript in line with his comments. The changes are listed below. Let us here just briefly comment on two points:

  • With regards to point 2 by the Referee, he is right in pointing that the eigenstates should generally be associated with a series of quantum numbers, one for each charge (Hamiltonian and other conserved charges). With this in mind, we have replaced |n> with | n, i_1, ... i_{N_cons} >, and indicate explicitly “n stands for the quantum number that identifies the energy eigenvalue, E_n, while i_k is the quantum number labelling the eigenvalues, M_{k, i_k }, of the charge operator \hat{M}_k.”, and similarly for the final states |m’> (lines 104-108).

  • On the Referee's comment #3: Our generalised quantum fluctuation relations (as is generally the case with these kind of equalities) are valid for any time dependence in H(t). This generality includes processes where H(t) is parameterised by a continuous parameter, say H = H(lambda(t)) with lambda(t) a continuous function of t. But the QFRs can be also applied to quantum quenches, i.e., sudden changes of the parameters of the Hamiltonian. This is the case we have analysed in our simulations where the parameters (alpha, g) are changed in a step-wise manner. We now provide explicit details on this point when describing our numerical protocol in Section 3.2, and have added a more general statement in Sec. 2 (step 2 of the generalised TPM protocol).

We hope that, with these changes, our manuscript will be accepted for publication in SciPost Physics Proceedings.

---

## Round 3 · List of Changes

1. We have added Refs. [1-4] to address the Referee’s comment #1.

2.In reply to the Referee’s comment #2: We have clarified the notation of the initial and final eigenstates, to account for the various conserved charges at each stage of the protocol.

3. In reply to the Referee’s comment #3: We provide a more detailed description of the time dependence of our Hamiltonian, with a new sentence in Section 2 (step 2 of the generalised TPM protocol), and a new short paragraph including a new equation specifying the functions {alpha(t), g(t)} we used in Section 3 (step 2 of the numerical protocol).

4. We corrected the text as requested by the Referee’s comment #4. The set of references [22, 34-36, 38] at the end of that sentence contains the further details on the existence of the additional conserved charge for the various values of alpha. We have also removed the subindex of M_k in line 152: as there is a single charge, there is no need to add a counter to distinguish it from others as in the general case.

---

## Editorial Decision

published